# Climate-Related Distribution Shifts of Migratory Songbirds and Sciurids in the White Mountain National Forest

**Aimee Van Tatenhove** [1,*], **Emily Filiberti** [1], **T. Scott Sillett** [1], **Nicholas Rodenhouse** [2] **and Michael Hallworth** [1]

[1]  Migratory Bird Center, Smithsonian Conservation Biology Institute, Washington, DC 20008, USA; efiliberti@une.edu (E.F.); silletts@si.edu (T.S.S.); mhallworth@gmail.com (M.H.)

[2]  Department of Biological Sciences, Wellesley College, 106 Central Street, Wellesley, MA 02481, USA; nrodenho@wellesley.edu

[*]  Correspondence: aimee.van.tatenhove@aggiemail.usu.edu; Tel.: +01-715-529-0159

**Abstract:** Climate change has been linked to distribution shifts and population declines of numerous animal and plant species, particularly in montane ecosystems. The majority of studies suggest both that low-elevation avian and small mammal species are shifting up in elevation and that high-elevation avian communities are either shifting further upslope or relocating completely with an increase in average local temperatures. However, recent research suggests numerous high elevation montane species are either not shifting or are shifting down in elevation despite the local increasing temperature trends, perhaps as a result of the increased precipitation at high elevations. In this study, we examine common vertebrate species distributions across the Hubbard Brook valley in the White Mountain National Forest, including resident and migratory songbirds and small mammals, in relation to historic spring temperature and precipitation. We found no directional change in distributions through time for any of the species. However, we show that the majority of low-elevation bird species in our study area respond to warm spring temperatures by shifting upslope. All bird species that shifted were long-distance migrants. Each low-elevation migrant species responded differently to warm spring temperatures, through upslope distribution expansion, downslope distribution contraction, or total distribution shift upslope. In contrast, we found a majority of high-elevation bird species and both high- and low-elevation mammal species did not shift in response to spring temperature or precipitation and may be subject to more complex climate trends. The heterogeneous response to climate change highlights the need for more comprehensive studies on the subject and careful consideration for appropriate species and habitat management plans in northeastern montane regions.

**Keywords:** climate change; temperature; precipitation; Hubbard Brook; elevational shifts; mountains

## 1. Introduction

Mounting evidence suggests plants and animals are responding to ongoing climate change in numerous ways, including through significant distributional shifts [1–3]. A simple paradigm asserts that species will respond to rising temperatures by shifting their distributions poleward (e.g., References [4–6]) or up in elevation [2,3,7]. However, as time goes on, studies are finding some species are either not shifting or are shifting downslope through time, even as temperatures increase [8–11]. Yet, it is not clear what is driving montane species to shift heterogeneously and how montane species are ultimately moving in the face of climate change.

Montane species can be particularly susceptible to the effects of climate change, because their distributions are elevationally constrained, leaving them vulnerable to range restriction or local extirpation through an inability to react sufficiently to a changing climate [12–14]. Montane songbird species and some small mammal species often occupy narrow niches [15,16] and therefore may be sensitive to climate induced habitat alteration [17,18]. The elevational gradient of montane regions may also exacerbate the severity of climate change, as climate regimes are not changing uniformly at all elevations [19–21]. Most notably, precipitation rates are typically greater at high elevations, and precipitation is expected to increase more rapidly at high elevations with ongoing climate change [19–21]. As a result, precipitation may affect high-elevation species disproportionately [10,21], and in some cases, species may shift down to find areas with more favorable precipitation [12,21]. Temperature may also increase more rapidly at some higher elevation sites [20,22], putting further stress on montane communities. Migratory bird species may be particularly impacted, as individuals must recolonize areas yearly and may therefore shift greater distances than non-migratory species [23]. Songbirds and small mammals in northeastern montane forests may be especially vulnerable, as these regions are highly threatened by climate change [19,20]. However, with few studies of climate induced distributional shifts in northeastern forests, it is relatively unknown how climate change will impact songbird and small mammal species in these regions.

Long-term studies of songbird and small mammal communities are crucial to understanding how these communities are responding to a changing climate. The songbird and small mammal communities have been systematically surveyed annually since 1999 at the Hubbard Brook Experimental Forest (HBEF) within the White Mountain National Forest. Here, we used these long-term survey data to test whether changing climate has affected songbird and small mammal distributions in northeastern hardwood and boreal forests. We hypothesized that both temperature and precipitation contribute to fine scale distributional shifts within the songbird and small mammal communities and that songbird migratory status would affect the magnitude of the distribution shifts. If songbird and small mammal distributions are governed by climate, we predicted low-elevation species would shift upslope with warm temperatures, while high-elevation species would predominantly shift downslope in response to increased high-elevation precipitations. Additionally, we predicted that migratory songbird species that recolonize breeding grounds within HBEF yearly would exhibit larger distribution shifts than resident species.

## 2. Materials and Methods

### 2.1. Study Site

Hubbard Brook Experimental Forest (43º56′N, 71º45′W, NH, USA; HBEF) is a 3600 ha watershed located within the White Mountain National Forest, ranging from 200–1000 m above sea-level (Figure 1). HBEF is forested and comprised of northern hardwoods found predominantly at low to middle elevations, transitioning to boreal spruce-fir forests at elevations above 800 m [24]. The climate within HBEF is temperate, with long, cold winters and mild, wet summers [25]. Over 100 bird species regularly breed within HBEF, the majority of which are Neotropical migrants that spend temperate winters in the tropics. Eastern chipmunks (*Tamias striatus* Linnaeus; EACH) and red squirrels (*Tamiasciurus hudsonicus* Erxleben; RESQ) are also common in the valley and are frequent nest predators of these bird species [26,27].

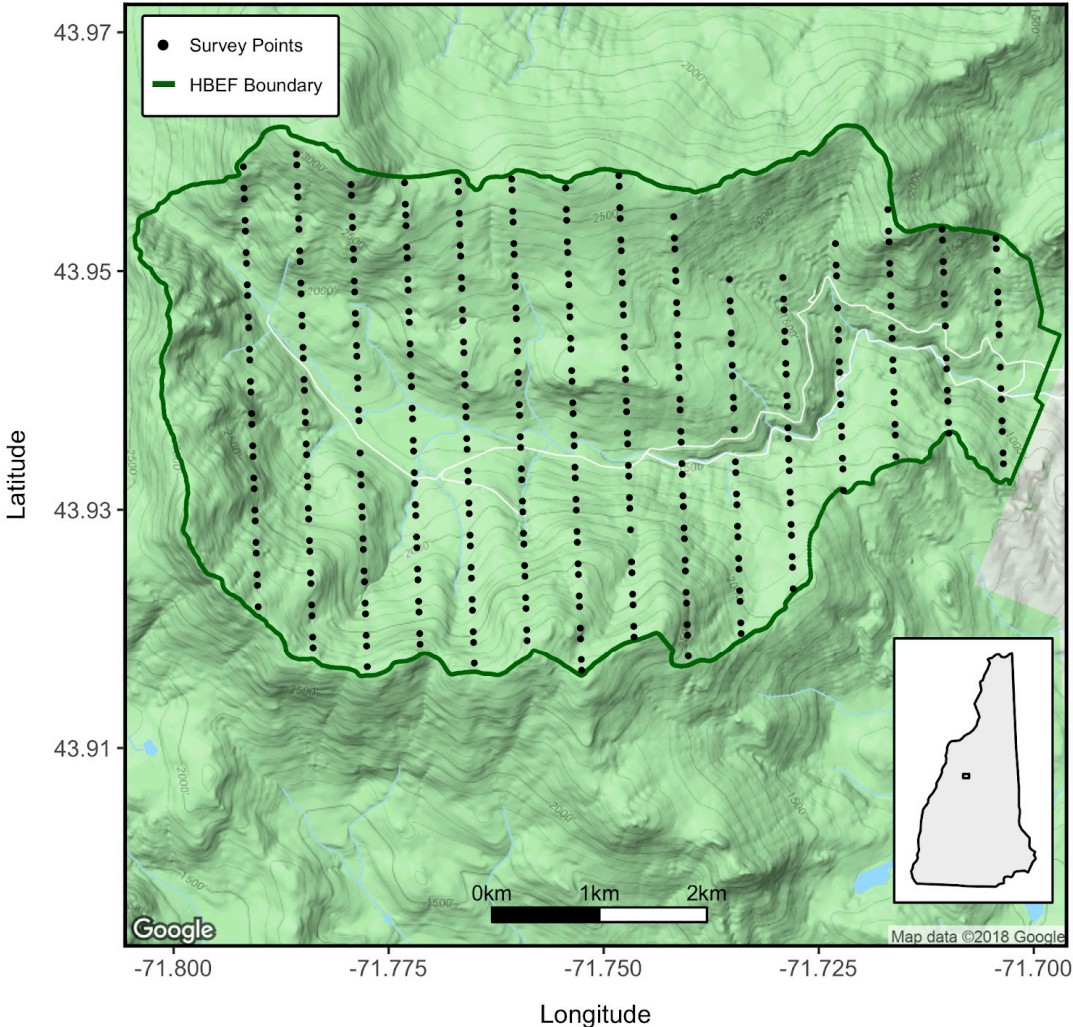

**Figure 1.** Map of Hubbard Brook Experimental Forest (HBEF) with survey locations in relation to New Hampshire.

## 2.2. Survey Methods

Point count surveys were used to collect avian and mammal occupancy data and were conducted annually from 1999 to 2016, excluding 2003 and 2004. Counts were conducted along 15 north–south transects, separated by 500 m that span the elevation gradient within HBEF. Survey locations along each transect were spaced either 100 or 200 m apart. Each survey location (*n* = 373) was surveyed at least three times during the breeding season (May through July), by a different trained surveyor each time. Point counts were conducted between 0530 and 1000 EDT (Eastern Daylight Time). Counts were not conducted in conditions that could hinder the surveyors' ability to detect individuals (rain, high winds, canopy drip, fog, etc.). During the ten-minute survey, all bird species, EACH, and RESQ seen or heard within 50 m of the point were recorded. Birds and mammals assessed to be outside 50 m were not used in this study to avoid accidental double counting of individuals at adjacent points.

## 2.3. Surveyed Species

We selected five songbird species to represent low-elevation bird species and five songbird species to represent high-elevation bird species, based on our prior knowledge of their breeding

distributions within HBEF and to remain consistent with species selected in other regional studies [12,13]. Species designations were then confirmed using species occupancy curves over the elevation gradient in our study site, as outlined in the statistical methods. The low-elevation species we selected were the black-capped chickadee (*Poecile atricapillus* Linnaeus; BCCH), black-throated blue warbler (*Setophaga caerulescens* Gmelin; BTBW), hermit thrush (*Catharus guttatus* Pallas; HETH), ovenbird (*Seiurus aurocapilla* Linnaeus; OVEN), and red-eyed vireo (*Vireo olivaceus* Linnaeus; REVI). The high-elevation species we selected were the blackpoll warbler (*Setophaga striata* Forster; BLPW), magnolia warbler (*Setophaga magnolia* Wilson; MAWA), dark-eyed junco (*Junco hyemalis* Linnaeus; DEJU), Swainson's thrush (*Catharus ustulatus* Nuttall; SWTH), and winter wren (*Troglodytes hiemalis* Viellot; WIWR).

Red squirrel (RESQ) data were available for all years, excluding the two years during which surveys were not conducted, 2003 and 2004. The eastern chipmunk (EACH) data was missing one additional year of data (2002). No other mammals were surveyed systematically during our study period. We categorized mammal species into low- and high-elevations using the same criteria we used for birds. EACH were designated as a low-elevation species and RESQ as a high-elevation species because of their association with conifers [16].

## 2.4. Environmental Variables

For this study, we focused on climate variables with long-term datasets available through our study period (1999–2016). We were primarily interested in how temperature and precipitation during the months when surveys were conducted affected species distributions within HBEF. Most migratory species arrive back at HBEF to breed in May. Therefore, we used mean May temperature and precipitation as potential drivers of distribution shifts since they coincide with the arrival of migratory bird species [28] and when RESQ and EACH pups first typically become active [29,30] (hereafter called "spring temperature" and "spring precipitation", unless specified). For migratory bird species that arrive at the breeding grounds in late May, and as such may not be influenced by early May weather, we chose to test whether mean June temperature and precipitation influenced their distribution shifts. Daily temperature and precipitation data from 1999 to 2014 were downloaded from the Long Term Ecological Research Network website (https://portal.lternet.edu). Additional temperature data for 2015 and 2016 were provided by the US Forest Service. All temperature and precipitation data were collected from the weather station at the USDA Forest Service Headquarters building at HBEF (252 m above sea level). Daily mean temperature and precipitation were averaged each year over May and June to generate mean annual May and June temperature and precipitation values.

## 2.5. Statistical Methods

Imperfect detection during animal surveys can lead to biased occupancy and abundance estimates and is common in point count data [31]. To account for imperfect species detection in our analyses, we used single and multi-season occupancy models (*unmarked package* v0.12-0, [32]) to predict true species occupancy in relation to elevation. Using these adjusted occupancy estimates, we confirmed our a priori songbird and small mammal elevation designations by assessing how their multiyear site occupancy probability varied over the elevation gradient within HBEF (Figure 2). Species occupancy curves that increased to 100% predicted the occupancy above 800 m (approximately where the ecotone between deciduous and boreal forests reside in HBEF) were confirmed as high-elevation species, and species with a predicted occupancy that peaked and then declined before 800 m in elevation were confirmed as low-elevation species.

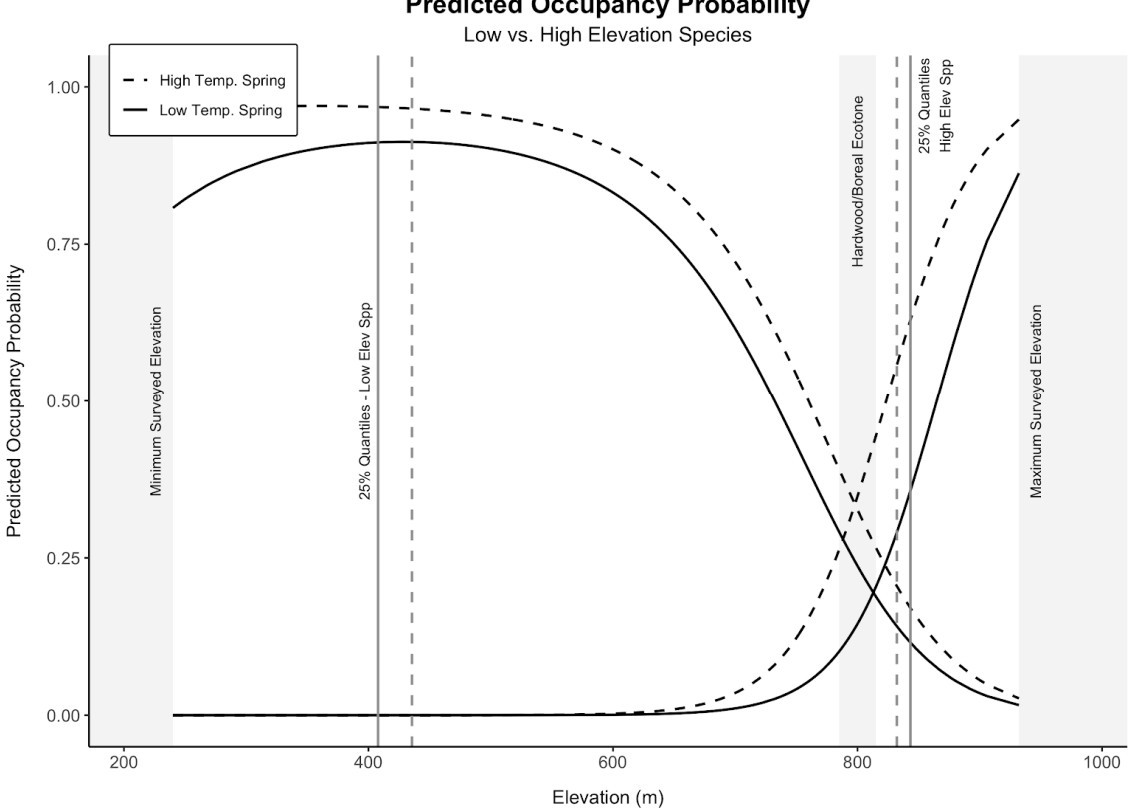

**Figure 2.** The simulated occupancy probabilities of high- and low-elevation species occupying a given elevation within HBEF, New Hampshire in relation to varying mean spring temperatures based on survey data corrected for imperfect detection. The solid lines represent species occupancy in a year with low spring temperatures, and the dashed lines represent occupancy in a year with high spring temperatures. The quantiles show how species distribution changes within their yearly range with mean spring temperatures.

The elevational gradient within HBEF (approximately 200 to 1000 m) likely does not encompass the full elevational distribution of some of the species included in our analyses, particularly those of high-elevation species. Following DeLuca and King [12], we accounted for partial elevational distributions by segmenting the predicted occupancy for each species into quantiles (2.5%, 5%, 25%, 50%, 75%, 95%, and 97.5%) to allow us to assess how species distributions were shifting and whether species distributions shifted uniformly, contracted, or expanded. Separate linear models for each predicted occupancy quantile were used to assess whether elevational distributions shifted through time within our study period. Linear models for each quantile were then used to assess whether the observed elevational distribution shifts were in response to average annual spring temperatures or average annual spring precipitations. We used Akaike's Information Criterion for limited sample sizes (AICc, MuMIn package v1.42.1, [33]) to identify the most parsimonious models among our candidate set of models. For low-elevation bird species, we examined the relationship between environmental covariates and the elevation of the 97.5% quantile across all low-elevation species, as the upper distributional range appears more sensitive to changes in climate [2,21]. For high-elevation bird species, we focused on the 2.5% quantile across all high-elevation species because HBEF is at the lower elevation band of their distribution. For our mammal species, we tested model appropriateness at the 50% quantile across both mammal species because our mammals include both high-elevation and low-elevation species.

All data are presented as mean ± 95% confidence interval, and results were considered significant at $p < 0.05$. All data were analyzed in R (v3.5.1, [34]).

## 3. Results

### 3.1. Environmental Variables

Historic mean annual temperature (df = 59, $R^2$ = 0.36, $p$ < 0.001; Figure 3) and precipitation rates (df = 35, $R^2$ = 0.09, $p$ = 0.044; Figure 3) have increased within HBEF, mirroring the trends observed in the northeastern North America over the past 100 years [19,20]. However, there was no change in the mean May temperature (df = 16, $R^2$ = 0.06, $p$ = 0.165) or precipitation (df = 14, $R^2$ = -0.07, $p$ = 0.909) over the 18-year period (1999 to 2016) that coincides with our survey data. The same 18-year period had substantial annual variability in both temperature and precipitation (Figure 3).

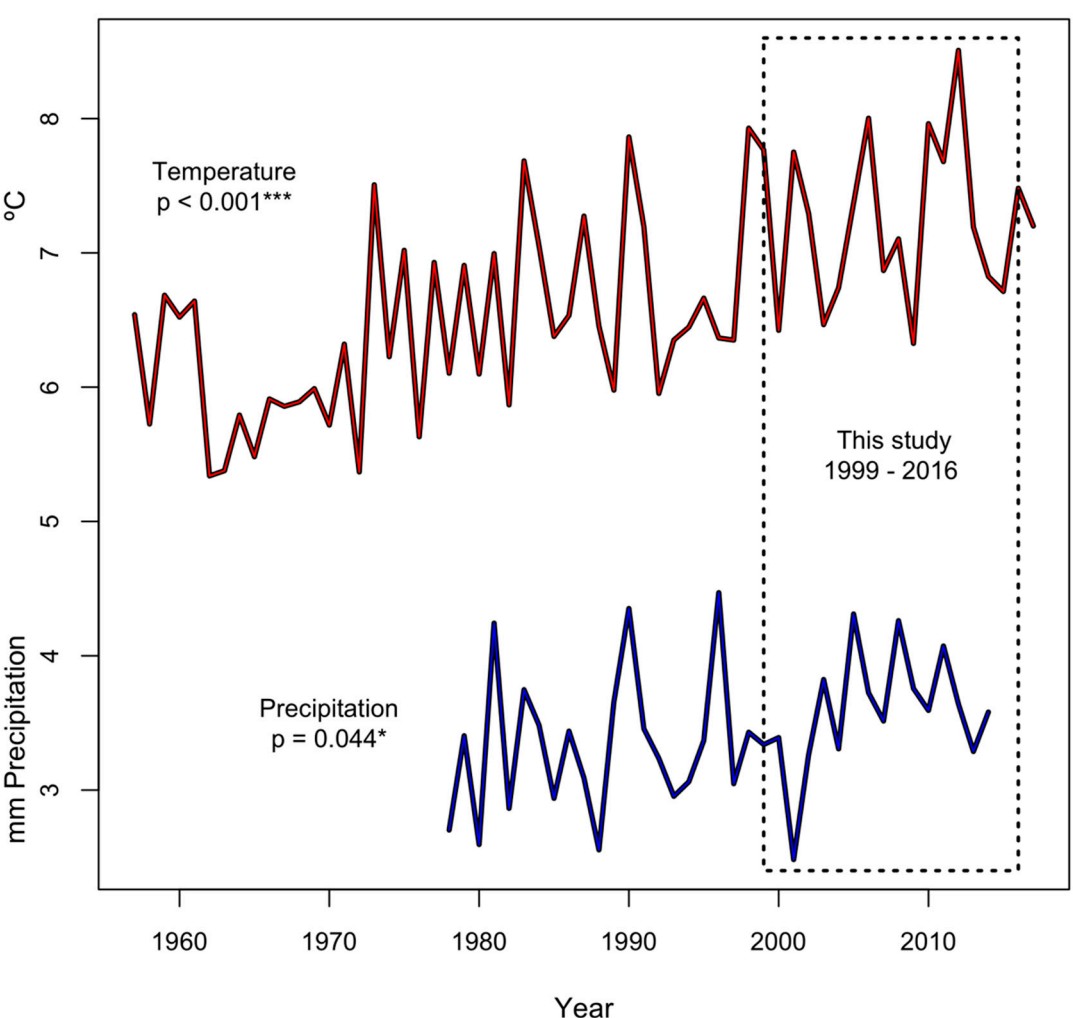

**Figure 3.** The mean annual temperature (1957–2017) and precipitation (1978–2014) trends in HBEF, New Hampshire. All data were collected from the USDA Forest Service Headquarters building at HBEF at 252 m above sea level.

### 3.2. Low-Elevation Birds

We found no evidence that low-elevation songbird species distributions shifted significantly with time over the study period (Table S1). However, three of five low-elevation bird species shifted upslope significantly with warm spring temperatures (BTBW, OVEN, and REVI; Table 1 and Table S2), while no-low elevation species shifted as a result of mean spring precipitation (Table S3). BTBWs contracted the bottom half of their distribution upslope by an average of 9.11 m/°C, while their upper

distribution above 50% remained stable. In contrast to BTBWs, OVENs expanded their upper distribution (95 to 97.5% quantiles) upslope by an average of 10.58 m/°C, while their lower distribution remained stable. REVIs shifted nearly their entire elevational distribution (5% to 97.5% predicted occupancy quantiles) upslope significantly with warm spring temperature, moving the top 50% of their distribution upslope an average of 16.94 m/°C, with the top 5% shifting an average of 20.48 m/°C. All (100%) of the species that shifted significantly were species that migrate to wintering grounds outside of the United States [35–37] (hereafter "long-distance migrants"), while the remaining two species were residents or migrants within the United States [38,39] (hereafter "short-distance migrants").

The mean spring temperature alone was the most parsimonious of our candidate models for low-elevation bird species (Table S4), although two other models (temperature + precipitation and time + temperature) had ΔAICc values of <2.0.

**Table 1.** HBEF songbird and small mammal species distribution shifts in relation to the mean spring temperature. The mean May temperature was used for all species, excluding BLPW, where the mean June temperature was used. Asterisks (*) denote statistically significant *p*-values ($p < 0.05$), and periods (.) denote near significant *p*-values ($0.05 < p < 0.10$). Abbreviations: Black-capped chickadee (BCCH), black-throated blue warbler (BTBW), hermit thrush (HETH), ovenbird (OVEN), red-eyed vireo (REVI), blackpoll warbler (BLPW), magnolia warbler (MAWA), dark-eyed junco (DEJU), Swainson's thrush (SWTH), winter wren (WIWR), red squirrel (RESQ), and eastern chipmunk (EACH).

| Mean Spring Temperature (1999–2016) | | | | | |
|---|---|---|---|---|---|
| **Low-Elevation Species** | | | | | |
| Spp. | % Occu | Intercept | $R^2$ | *p* | Significance |
| | 2.5% | 8.71 | 0.30 | 0.016 | * |
| | 5% | 10.29 | 0.38 | 0.007 | ** |
| | 25% | 10.40 | 0.61 | 0.000 | *** |
| BTBW | 50% | 7.06 | 0.38 | 0.007 | ** |
| | 75% | 3.77 | −0.01 | 0.370 | |
| | 95% | 3.43 | −0.04 | 0.545 | |
| | 97.5% | 4.39 | −0.03 | 0.446 | |
| | 2.5% | -0.90 | −0.01 | 0.390 | |
| | 5% | -1.00 | −0.04 | 0.534 | |
| | 25% | 0.40 | −0.07 | 0.882 | |
| OVEN | 50% | 2.78 | −0.01 | 0.361 | |
| | 75% | 5.78 | 0.10 | 0.129 | |
| | 95% | 10.12 | 0.22 | 0.040 | * |
| | 97.5% | 11.05 | 0.21 | 0.041 | * |
| | 2.5% | 0.77 | 0.14 | 0.082 | . |
| | 5% | 1.62 | 0.21 | 0.040 | * |
| | 25% | 6.43 | 0.53 | 0.001 | *** |
| REVI | 50% | 10.95 | 0.67 | 0.000 | *** |
| | 75% | 15.86 | 0.70 | 0.000 | *** |
| | 95% | 21.55 | 0.55 | 0.001 | *** |
| | 97.5% | 19.41 | 0.42 | 0.004 | ** |
| **High-Elevation Species** | | | | | |
| | 2.5% | 14.81 | −0.06 | 0.656 | |
| | 5% | 9.47 | −0.01 | 0.392 | |
| | 25% | 4.93 | 0.21 | 0.041 | * |
| BLPW | 50% | 3.47 | 0.34 | 0.011 | * |
| | 75% | 1.83 | 0.33 | 0.012 | * |
| | 95% | 0.42 | 0.47 | 0.002 | ** |
| | 97.5% | 0.16 | 0.23 | 0.034 | * |

### 3.3. High-Elevation Birds

We found that most high-elevation songbird species distributions did not shift significantly with time. Two species shifted portions of their distributions upslope with time (SWTH and DEJU) but only for one predicted occupancy quantile for each (25% and 75% respectively; Table S1). One of three

high-elevation, long-distance migrant species [40] shifted distributions with the mean spring temperature (BLPW; Table 1), while the remaining long-distance migrants [41,42] and resident species [43,44] did not (Table S2). Contrary to our predictions, no high-elevation species shifted downslope, with the exception of the BLPW. The upper edge of the BLPW's distribution, at the maximum elevation of our study site, contracted slightly downslope with warm May temperature (97.5% quantile; −0.14 m/°C). However, BLPW distribution from the 25% to 97.5% quantiles expanded upslope with the mean June temperature (Table 1), with an average upslope shift of 2.16 m/°C.

No high-elevation bird distributions shifted as a result of the mean spring precipitation (Table S3). Our most parsimonious model of our candidate high-elevation species models was spring precipitation (Table S4), although models for both temperature and time had ΔAICc values of <2.0.

*3.4. Mammals*

We found no evidence that time, mean spring temperature, or mean spring precipitation had any significant effect on RESQ or EACH distributions through our study period (Table S1–S3). The mean spring temperature was the most parsimonious model of our candidate models for mammals (Table S4) but only slightly more so than precipitation (ΔAICc = 0.05) and time (ΔAICc = 0.11).

## 4. Discussion

Overall, we did not find that songbird or small mammal elevational distributions were shifting through time within our study period. This is perhaps because temperature and precipitation did not increase significantly over our study period. Instead, our study species responded proximately to annual variations in climatic variables, with the majority of low-elevation songbird species responding to warm spring temperatures by shifting upslope (Figure S1a). However, most high-elevation songbird species and both small mammal species did not respond to temperature or precipitation (Figure S1a,b), perhaps because shifts seen in other studies occurred at higher elevations than those within our study site and climate constraints may be more extreme at higher elevations [8,11,12].

The distributional shifts we observed in low-elevation songbirds may be related to migratory status. Avian migratory status has been assessed by a handful of other studies (e.g. References [21,23]) but is not commonly assessed by studies looking at distributional shifts. Walther et al. [23] argue that migratory species are more likely to shift than resident species because they recolonize areas yearly. Within HBEF, first time breeders often must colonize new territories and are therefore subject to distribution shifts through changes in abundance and reactions to climatic variables and territory quality. This is in contrast with the resident species that take the slower, more stable route of local population extinction and colonization. They therefore exhibit smaller and slower distribution shifts [23], and is perhaps why we did not see shifts in resident species. Similar to Walther's [23] findings, we found that all three of our low-elevation songbird species that shifted upslope with warm spring temperatures were long-distance migrants, while the two species that did not shift significantly were a resident species (BCCH) and a short-distance migrant (HETH). Notably, our low-elevation migrant with the longest migration route and latest spring arrival time of our low-elevation songbirds, the REVI, was our most responsive species to warm spring temperatures. REVIs shifted nearly their entire distribution upslope with warm spring temperatures and shifted upslope farther than any of our other low-elevation species, at almost +20 m per degree of temperature increase. The other two low-elevation species that responded to warm spring temperature (BTBW and OVEN) responded to a lesser degree, shifting shorter distances and only portions of their distribution (Table 1). Long distance migrants have no way of assessing specific climate conditions at their summer breeding grounds, and they may instead use cues other than temperature, including day length [45,46] and favorable flying conditions at wintering grounds [47] to determine when to begin spring migration. Plastic responses to habitat quality upon arrival at summer breeding grounds may help buffer against phenological mismatches inherent with having a relatively set migratory schedule. Species with the longest migration routes, such as the REVI, which winters in South America [37], may benefit from

having plastic settlement patterns, and these long migration routes may be one of the underlying reasons we see REVIs shift more than other species.

Contrary to our predictions, low-elevation mammals did not shift significantly with time, temperature, or precipitation. Although EACH is a frequent nest predator of songbird eggs and nestlings [26,27], their diet consists primarily of tree seeds, including those from the exceptionally long-lived [48] American beech trees (*Fagus grandifolia* Ehrhart) and sugar maple trees (*Acer saccharum* Marshall) [29,49]. EACH are therefore highly reliant on seed producing tree species as a food source, especially for caching food for consumption over the winter months [29,49]. In non-mast years, when tree seeds are scarce, EACH still rely heavily on vegetative food sources [29]. This reliance on seed producing tree species and other vegetation may be driving distribution shifts, or the lack thereof, as EACH may be tracking the slower changes in distribution of long-lived tree species over changes in temperature and precipitation [50].

In contrast with low-elevation songbird species, we found no distributional shifts in high-elevation songbirds (Figure S1a), with the exception of the BLPW, which shifted downslope with warm May temperatures. However, with an average shift of −0.14 m/°C, BLPW distributional shifts as a result of the mean May temperature were much smaller than those seen in low-elevation species. Others have also documented high-elevation birds shifting downslope [8,12,21], although the proposed causes vary. Some potential explanations include shifting vegetation [8,12] and increased high-elevation precipitation [12,21]. Neither of these explanations appear to be occurring within HBEF, as HBEF forest structure appears stable [51] and BLPWs did not respond to the mean spring precipitation. However, BLPW distributional shifts may be responding to the mean June temperature. Contrary to shifts related to the May temperatures, BLPWs showed a significant upslope expansion through all but the lowest reaches of their distribution with warm June temperatures (Figure S1a). BLPWs are long-distance migrants [40] and are typically the latest birds to arrive in HBEF out of the species we assessed in this study. They typically arrive to HBEF near the end of May and begin nesting shortly after [40]. As a result, they are likely influenced most by the temperature at the end of May, which, within HBEF, are more typical of temperatures we would expect to see in June. So, why is the BLPW shifting while other high-elevation species are not? Like the REVI, BLPWs are one of the longest distance migrants of our high-elevation species [40] and may be more plastic in their response to environment as a way to buffer against phenological mismatches when arriving at breeding grounds.

Overall, the lack of observed distribution shifts by high-elevation songbird and mammal species at HBEF, with the exception of BLPWs, may be due in part to the elevation range in our study site. Most of our high-elevation sites are lower than those in numerous other studies that observed birds and small mammals move downslope (elevational maximums: this study, 903 m; Archaux [8], 3099 m; DeLuca and King [12], 1470 m; etc.), and climate constraints on our high-elevation species may not be as extreme as those at higher elevation sites [19,20]. Boreal habitats within HBEF are found primarily above 800 m in elevation and are therefore restricted to the highest ridges in the valley. As a result, we also may be missing reactions to precipitation by high-elevation species due to the small elevational range of boreal spruce-fir within HBEF. Alternately, high-elevation species in HBEF may be experiencing the effects of increased precipitation but may not be shifting significantly out of necessity for specific breeding habitats. Archaux [8] found that avian abundance and distribution within two study sites in the French Alps was closely tied with habitat distribution shifts, despite significant warming. Similar studies of high-elevation mammals like RESQ have found they may also be tracking habitat distribution shifts over changes in climatic conditions, much like their low-elevation counterparts [11,50]. RESQ are heavily reliant on conifer seeds as a food source and typically hold territories in or near stands of conifers [16,52]. Elevationally restricted habitat may cause increased competition by habitat-constrained species for breeding or food caching territory [12,15,29], and as a result, high-elevation species may be using all of the available habitat that suits their needs, regardless of quality. If this is the case, any evidence of habitat tracking would likely only be seen over longer time periods, as vegetation distribution shifts are typically slower than temperature change, due to the long lifespans of trees [1,20].

*4.1. Future Implications*

Distributional shifts will likely impact montane passerines and small mammals in several ways. With temperature and precipitation rates expected to increase faster at higher elevations, species shifting upslope may encounter novel combinations of climate and vegetation. Also, because some species are shifting while others are not, the avian community composition will likely change as a result, and novel species interactions may arise [53]. As birds shift in elevation, they are also likely to encounter non-avian species and habitats they have not interacted with before [53,54]. While these novel interactions may be beneficial for some species, it is unknown how this will ultimately affect sensitive populations and ecosystem dynamics. For instance, nest predation is a major factor in avian reproductive success or failure [27,49,55], and as bird species distributions shift, birds moving into areas where RESQ and EACH overlap may encounter higher rates of nest predation, potentially impacting their ability to produce young. Additionally, species that are not shifting may encounter southern and lower-elevation species that are expanding into areas left by species moving upslope or poleward [6,13], like red-bellied woodpeckers (*Melanerpes carolinus*), tufted titmice (*Baeolophus bicolor*), and eastern gray squirrels (*Sciurus carolinensis*). Species with contracting distributions, like the BTBW, may also face increased intraspecies competition during the breeding season, as the preferred breeding habitat decreases.

Small yearly distributional shifts may not be ecologically significant alone, and over short timescales, the impacts of elevational shifts may not be obvious. However, as temperatures increase over the long term, these small shifts may lead to larger scale shifts that will affect ecosystems and individual species alike. The mean annual temperature in New England is expected to increase 1.7 to 4.4 °C or more by the year 2100 [19,20], and under the worst-case warming scenario, we could see REVI and other highly plastic species shift upslope an average of almost 100 m by the year 2100. For species occupying sites near the maximum elevation within HBEF, an upslope shift of 100 m will likely extirpate them from the valley entirely. Precipitation and temperature increases are expected to be more severe at higher elevations [20,21]. As a result, species at higher elevations in the region may shift even farther than the species detailed here [12,21,54], thus exacerbating the effects of elevational shifts and novel community interactions.

*4.2. Further Research*

Long-term datasets are invaluable for measuring species distribution shifts. Therefore, the continued collection of species occupancy and abundance coupled with climate variables is essential to understanding the impacts of climate change on montane species as well as impacts on lower elevation species that may eventually colonize montane habitats. It is likely other local and regional variables in addition to temperature and precipitation influence elevational distribution shifts in our study species. Other variables should be assessed to increase our understanding of how species distributions are changing, including vegetation shifts at transition zones, changes in winter snowpack depth, intra and interspecies interactions, and prey distribution shifts.

HBEF supports a wide variety of avian species, including high-elevation species, despite HBEF being at the lower end of many of their elevational distributions. Yet, as climate continues to change and species continue to shift upslope, we may begin to lose high-elevation species within HBEF as they move to higher elevation sites elsewhere. Mountaintop extirpation as a result of climate change has been the focus of only a few studies in the northeast and should be explored further to increase our understanding of how species distributions shift with climate change.

## 5. Conclusions

We found almost no directional change in distributions through time for any of our songbird or small mammal species. However, we found three of our five (60%) low-elevation bird species (BTBW, OVEN, and REVI) responded to warm spring temperatures by shifting upslope. All low-elevation songbird species that shifted were long-distance migrants, while those that did not shift were year-round residents of HBEF or short-distance migrants (BCCH and HETH, respectively). Each low-

elevation migrant species responded differently to warm spring temperatures, through upslope distribution expansion, downslope distribution contraction, or total distribution shift upslope. BLPW was the only high-elevation songbird species that shifted with warm spring temperature (downslope with the mean May temperature and upslope with the mean June temperature). The remaining high-elevation bird species (MAWA, DEJU, SWTH, and WIWR) and both high- and low-elevation mammal species (RESQ and EACH, respectively) did not shift in response to spring temperature or precipitation. This may be due to the limited elevation range in our study site, which may not experience the severe climate constraints found at higher elevations.

**Supplementary Materials:** The following are available online at www.mdpi.com/1999-4907/10/2/84/s1, Figure S1: Elevational Shifts in Target Species, Table S1: Hubbard Brook Experimental Forest (HBEF) songbird and small mammal species distribution shifts in relation to time in years, Table S2: HBEF songbird and small mammal species distribution shifts in relation to the mean spring temperature, Table S3: HBEF songbird and small mammal species distribution shifts in relation to the mean spring precipitation.

**Author Contributions:** Conceptualization, Aimee Van Tatenhove, Emily Filiberti, and Michael Hallworth; methodology, Aimee Van Tatenhove, Emily Filiberti, and Michael Hallworth; software, Aimee Van Tatenhove and Michael Hallworth; validation, Michael Hallworth; formal analysis, Aimee Van Tatenhove and Michael Hallworth; investigation, Nicholas Rodenhouse and T. Scott Sillett; resources, Nicholas Rodenhouse and T. Scott Sillett; data curation, Michael Hallworth; writing—original draft preparation, Aimee Van Tatenhove and Michael Hallworth; writing—review and editing, Michael Hallworth, Emily Filiberti, T. Scott Sillett and Nicholas Rodenhouse; visualization, Aimee Van Tatenhove; supervision, Michael Hallworth; project administration, Nicholas Rodenhouse, T. Scott Sillett and Michael Hallworth; funding acquisition, Nicholas Rodenhouse and T. Scott Sillett.

**Funding:** This manuscript is a contribution of the Hubbard Brook Ecosystem Study. Hubbard Brook is part of the Long-Term Ecological Research (LTER) network, which is supported by the U.S. National Science Foundation. The Hubbard Brook Experimental Forest is operated and maintained by the USDA Forest Service, Northern Research Station, Newtown Square, PA. This research was funded by the National Science Foundation, grant numbers 9810221 and 0423259.

**Acknowledgments:** A special thanks to Amey Bailey at the USDA Forest Service headquarters at HBEF for compiling missing years of temperature data on our behalf. We also thank our two anonymous reviewers whose helpful suggestions strengthened our manuscript considerably, Bill DeLuca for the helpful statistics suggestions, and all of the dedicated field technicians who collected data for our avian and mammal dataset.

**Conflicts of Interest:** The authors declare no conflict of interest.

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
