# Peer review of "Climate-Related Distribution Shifts of Migratory Songbirds and Sciurids in the White Mountain National Forest"

_forests, doi:10.3390/f10020084_

Round 1
Reviewer 1 Report
Minor edit: I assume the delta symbol will be corrected for the AICc.
It is a good, well written paper overall on a topic that is of interest. Having this long-term data set is valuable, but, I wonder why the authors were not able to have the sample size for enough species to make statistical comparisons (even non-parametrically) between migratory (long and short) and resident species - that would be very interesting. My second comment is about temperature and to a lesser extent precipitation data. I realize you can't change your monitoring site at 252m, but, that is near the beginning of your overall relief. Also, and somewhat related, using a measure of variance in temperature or precipitation may be more informative than means. Lastly, I would use 'squirrel' or sciurids in the title as 'small mammal' made me think there would be some Peromyscus, Insectivora, or other smaller small mammals.
Author Response
Attached are our responses to Reviewer 1. We appreciate the helpful comments, and we feel they improved the manuscript considerably.

Reviewer 2 Report
General Comments: This is a well written and robustly analyzed study that connects avian range shifts to specific climate variables. Such studies are an important step toward understanding how species will respond to climate change. I have broad issues that need to be addressed. The first is that I think their assertion that migratory behavior can influence the extent to which the species can shift is interesting and should be included more formally. See comments regarding this below. Second, I think Table 1 and some of the tables in the appendix should be switched and/or combined. Most of the interesting information is in the Appendix and I think the authors can do a better job of summarizing that info in the paper in either a table or figure. Further I think the analysis behind table 1 needs to be clarified. Figure 3, which is barely and incompletely referenced in the paper should be moved to the Appendix. See below for more specific comments. After these suggested changes and comments are addressed, I think this paper should be published.
Good clear abstract.
Title: The title implies that there are small mammal distribution shifts and as far as I could tell there are none. Also, the plural use of the word “forests” leaves the reader believing that this study occurred at several sites throughout the entire northern temperature forest. I don’t see anything wrong with being more specific to your area of inference here. Maybe just the HBEF? Or the White Mountains? Do you really comfortable inferring across the northern temperate forest based on findings from one 3,600ha watershed? I wouldn’t. Focusing in on one site like this is important. It typically allows you to dive in a bit deeper than you could if you were limited to climate or species data at broader spatial scales.
Line 41: I think the current paradigm is already shifting so I’m not sure I agree with this statement. This sounds more like the original paradigm or a simple paradigm. I think your set up in the first paragraph should center on your last sentence. Most studies look at changes over time or some loose correlation with a broad climate variable. Yours is unique in that it uses a specific climate variable that can help identify mechanisms.
Line 50: This sentence is odd. Clarify or remove.
Line 53: This citation seems relevant here.
Seidel TM, Weihrauch DM, Kimbull KD, Pszenny AAP, Soboleski R,Crete E, Murray G (2009) Evidence of climate change declines with elevation based on temperature and snow records from 1930s to 2006 on Mount Washington, New Hampshire, USA. Arct Antarc Alp Res 41:362–372
Line 126: Given you have some negative results and a nice climate dataset, did you try looking at any extreme measure of these data? Min, max, total? I don’t think you need to go on a fishing trip but it seem plausible that extreme hot, cold, drought or excessive rain events could be important.
Line 137: I like this. You should refer to this process above when you define high and low elevation species.
Line 158: These extreme quantiles may have a small subset of actual data to inform them. I would recommend either telling the reader what the sample size is for each species in these distributional tails and/or also running this analysis for other quantile thresholds, at least 5%. Further, maybe I missed it, but it is unclear how you summarized across species for this analysis. Were all species pooled into 1 model? Were species included as a fixed or random effect? Its clear for the previous analysis you used a single species, multi-season models. Then on line 153 you say you use linear models, but don’t address how you have 1 set of candidate models for all low elevation species. This needs to be clarified. I also question what the AIC comparison (Table 1) adds to this study? The real information is in the individual species models in Table S1. I would rather see a summarized version of the supplemental tables than your current Table 1, it seems a bit redundant and with less info than the species analysis. If this is not the case, you should do a better job of justifying the inclusion of this analysis above and beyond the species-based analysis. I do understand to the need to provide more information when trying to prove the lack of an effect. However I think there is much more info in your supplemental tables and at the very least you should switch the existing main table with the supplemental tables. I realize they are quite large, maybe you can figure out a way to summarize them in a smaller table or figure for the main paper and then keep the longer tables as supporting info? Another possibility is to use your
Fig 2. This is super blurry in the pdf. Very hard to read the text embedded in the graph. Bummer. Why not say what species these are?
Table S1: I think it is great that you report the stats for each %Occu. Given we don’t know which cutoff accurately represents meaningful distributional measures, seeing them all makes sense.
Figure S1: Unless the journal has specific restrictions I think you should include this figure in the main body of the paper. It’s important to set the stage that climate in HBEF is dynamic over your study period.
Line 196: Many readers will not look at the supplement. Where you can, include as much info in the main body of the paper. Here, just say BLPW was the only high elevation species that had evidence of shifting with temp.
Line 200: For low elevation species you start with the comparison to shifts over time yet for high elevation birds you start with the comparison to climate. You should keep this consistent. This would also keep it consistent with the order of your tables in the supplement.
Line 202: Now you are re-report what you already said on line 146. Please go through the high elevation results section and improve its organization.
Line 203: I think you should be careful with how you use the word “range”. I know you explain how you dealt statistically with this, but this is really just your sampled range which likely is a small portion of its elevational distribution.
Figure 3: If you only reference Fig 3a, maybe you only need to show that panel. It also seems a bit odd to only reference this very result based figure only in the discussion.
Discussion: How does the elevational range of your data compare to that of other studies in the region that found shifts, particularly for high elevation species. Is it possible that shifts are occurring but generally only seen at elevations higher than you sampled? If so you should say as much.
Line 221: I think this is a great question. Why not raise it in the intro as one of your objectives/hypotheses? Then you summarize your finding related to this question in your results section and discuss an actual result of your work. Here you are spending the second paragraph of your discussion on a finding that is not presented in your results section. I could be something as simple as a sentence reporting %migrants with a shift. More importantly it should be an explicate objective or hypothesis in the intro.
Line 226: what about a graph plotting migration distance to shift coefficient? Just an idea, but could be a nice way to show this.
Line 239: What about an explanation of why you didn’t see a shift in residents?
Figure 3: Also very blurry and hard to read. The fact that you haven’t referred to this figure until the middle of the discussion and that you only refer to panel A is telling that as the paper currently stands, this figure is not necessary. You should either do a better job of justifying the inclusion of this figure by appropriately referring to it more often or move it to the supplement.
Line 247: I know there are a lot of habitat manipulations that occur within HBEF, what role could that play in your study if any?
Line 248: This statement begs the question, given you have the data, can June temp explain this? If you aren’t willing to explore this maybe don’t make this statement since it appears you have the data to explore this further.
Line 251: Really? Even later than SWTH?
Line 257: This really seems like an interesting thread and again, I would encourage you make more of an explicit component of the paper by introducing it in the intro and not just as an aside in the discussion.
Line 263: This is what I was looking for earlier. You might want to mention this earlier and let the reader know it will be discussed in more detail later. As a reader, this issue has been nagging me for a good chunk of the paper.
Line 267: Are all of these relevant? I think only the studies of similar latitude and mountain structure are comparable to your study.
Line 283: I’m surprised there isn’t a discussion of your mammal results before the future implication section. Even if its short, it seems like its an oversight to essentially not mention it. The mammal analysis currently seems like a forced “and” at the end of many sentences. If you keep it in the paper, at least discuss the results.
Line 313: Most of HBEF isn’t montane, don’t you think its import to collect these data for low elevation species too, which might ultimately montane?
Line 317: What about population fluctuations? Can increasing populations in the region explain the expansions you observed? Can declines explain contractions? I think this is important to discuss.
Author Response
Attached are our responses to Reviewer 2. We appreciate the helpful comments, and we feel they improved the manuscript considerably.
